# A unified framework for inferring the multi-scale organization of chromatin domains from Hi-C

**Ji Hyun Bak**[1☯¤], **Min Hyeok Kim**[1☯], **Lei Liu**[1,2], **Changbong Hyeon**[1,3]*

**1** Korea Institute for Advanced Study, Seoul, Korea, **2** Department of Physics, Zhejiang Sci-Tech University, Hangzhou, China, **3** Center for Artificial Intelligence and Natural Sciences, Korea Institute for Advanced Study, Seoul, Korea

☯ These authors contributed equally to this work.
¤ Current address: University of California, San Francisco, California, United States of America
* hyeoncb@kias.re.kr

**Data Availability Statement:** All codes are available from https://github.com/multi-cd. All data used in the paper were obtained from publicly available repositories. The following data were obtained through the NCBI GEO database: Hi-C

## Abstract

Chromosomes are giant chain molecules organized into an ensemble of three-dimensional structures characterized with its genomic state and the corresponding biological functions. Despite the strong cell-to-cell heterogeneity, the cell-type specific pattern demonstrated in high-throughput chromosome conformation capture (Hi-C) data hints at a valuable link between structure and function, which makes inference of chromatin domains (CDs) from the pattern of Hi-C a central problem in genome research. Here we present a unified method for analyzing Hi-C data to determine spatial organization of CDs over multiple genomic scales. By applying statistical physics-based clustering analysis to a polymer physics model of the chromosome, our method identifies the CDs that best represent the global pattern of correlation manifested in Hi-C. The multi-scale intra-chromosomal structures compared across different cell types uncover the principles underlying the multi-scale organization of chromatin chain: (i) Sub-TADs, TADs, and meta-TADs constitute a robust hierarchical structure. (ii) The assemblies of compartments and TAD-based domains are governed by different organizational principles. (iii) Sub-TADs are the common building blocks of chromosome architecture. Our physically principled interpretation and analysis of Hi-C not only offer an accurate and quantitative view of multi-scale chromatin organization but also help decipher its connections with genome function.

## Author summary

An array of square blocks and checkerboard patterns characteristic to Hi-C data reflects the multi-scale organization of the chromatin chain. Deciphering the structures of chromatin domains from Hi-C and associating them with genome function are open problems of great importance in genome research. However, most existing methods are specialized in finding domains at different scales, making it difficult to integrate the solutions. Here we develop a unified framework for modeling and inferring domain structures over

data, accession GSE63525; CTCF ChiP-seq, GSM749704; Repli-seq, GSM923451; histone signals, GSE29611; RNA-seq, GSE33480. RNA-seq data for the KBM7 cell were obtained from https://opendata.cemm.at/barlowlab. Information about genes (Known Genes table) and the associated regulatory elements (GeneHancer interaction table) were obtained from UCSC Genome Browser. The GeneCards database was consulted for individual gene information.

**Funding:** This work was supported in part by a KIAS Individual Grant at Korea Institute for Advanced Study (No. CG035003 to C.H.). The funders had no role in study design, data collection and analysis, decision to publish, or preparation of the manuscript.

**Competing interests:** The authors have declared that no competing interests exist.

multiple scales, based on a physical model of the chromosome that reflects its nature as a three-dimensional object. Our method efficiently explores the space of domain solutions at different genomic scales, and systematically infers the chromatin domains over multiple scales from Hi-C data by employing a single tuning parameter. Our principled interpretation of Hi-C not only offers a quantitative view of multi-scale chromatin organization but also helps understand its connections with genome function.

## Introduction

The spatial organization of chromatin inside cell nuclei has a profound impact on the function of the genome [1]. Chromosome conformation capture (3C) and its derivatives, which are used to identify chromatin contacts through the proximity ligation techniques [2, 3], take center stage in chromosome research [1, 4]. In particular, high-throughput chromosome conformation capture (Hi-C) technique quantifies all pairwise interactions between the segments of the chromatin. Square-block and checkerboard patterns that appear in Hi-C data provide glimpses into the organization of chromatin chains.

Despite a fundamental limitation of Hi-C that the data are in practice determined from a population of cells with strong cell-to-cell heterogeneity, the cell-type specificity and even the pathological states [5, 6] can clearly be discerned between different Hi-C maps. Given that Hi-C pattern changes with the transcription activity and along the phase of cell cycle [7–16], accurate characterization of chromatin domains (CDs) from Hi-C data is of great importance in advancing our quantitative understanding to the functional roles of chromosome structure in gene regulation.

Different types of CDs have been identified at different genomic scales. Inside cell nuclei each chromosome made of $\sim \mathcal{O}(10^2)$ Mb DNA is segregated into its own territory (Fig 1A) [17]. At the scale of $\gtrsim \mathcal{O}(10)$ Mb, alternating blocks of active and inactive chromatin are phase-separated into two megabase sized aggregates, called A- and B-compartments [18–21] (Fig 1B). Topologically associated domains (TADs), detected at $\sim \mathcal{O}(10^{-1}) - \mathcal{O}(1)$ Mb [22–25], are considered the basic functional unit of chromatin organization and gene regulation because of their well-conserved domain boundaries across cell/tissue types [17, 20, 26, 27]. It was suggested that the proximal TADs in genomic neighborhood aggregate into a higher-order structural domain termed "meta-TAD" [8]. At smaller genomic scale, each TAD is further split into sub-TADs that display more localized contacts [19, 28–31] (Fig 1C).

The current knowledge of CDs is based on a number of computational methods for Hi-C data analysis [18, 19, 22, 32]. Although each method made a unique and important contribution to the field, most are limited to finding CDs at specific scales under specific conditions. Lack of a systematic way of integrating or comparing these methods and their CD solutions engenders two issues. First, depending on the method being used, the identified CDs display significant variations, with the average size of a TAD varying from 100 kb to 2 Mb; yet there is no shared way to determine which size should be preferred. Second, it is difficult to formulate a comparative analysis between CDs found at different scales. To make things worse, many algorithms require that Hi-C data be pre-processed in specific ways, for example down-sampling to a coarser resolution for targeting CDs at larger scale [18, 19, 22]. Progress was made by methods that consider hierarchical CDs [33, 34]; however, these method fall short of providing a common interpretable framework, because they are restricted to local pattern recognitions, as in many earlier methods [18, 19, 22, 32]. To avoid the arbitrariness of focusing on the features at specific scales, a better approach will be to start from a physically principled notion

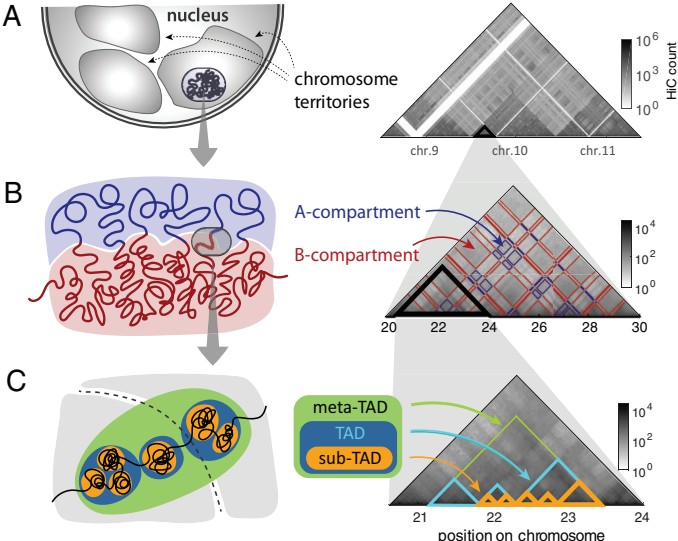

**Fig 1. The hierarchical organization of interphase chromosome and Hi-C map. (A)** Chromosome territories in the cell nucleus, which are manifested as the higher intra-chromosomal counts in the Hi-C map. **(B)** Alternating blocks of active and inactive chromatins, segregated into A- and B-compartments, give rise to the checkerboard pattern on Hi-C. **(C)** Sub-megabase to megabase sized chromatin folds into TADs. Adjacent TADs are merged to meta-TAD [8], and individual TAD is further decomposed into sub-TADs [19, 28–31].

that chromosomes are a hierarchically structured three dimensional object made of a long polymer [8, 16, 35–41].

Here we present a unified framework for characterizing CDs over multiple scales, incorporating an interpretable model of the chromosome based on polymer physics, and a clustering method based on statistical physics and Bayesian inference. For convenience, we will divide our framework to the two parts: *pre-processing* and *inference*. For pre-processing, Hi-C data is interpreted as a pairwise contact probability matrix, resulting from a Gaussian polymer network whose inter-monomer distances are restrained harmonically, which allows us to transform a raw Hi-C data into the correlation matrix. For the inference step, we formulate a statistical model of pairwise correlation with hidden domains, and set up an optimization problem to find the CD solution that can best explain the Hi-C-derived correlation matrix. The model flexibly explores the space of CD solutions at different genomic scales, and finds a family of solutions parameterized by a single parameter λ. We present a single method pipeline, Multi-CD, that combines the pre-processing and inference algorithms. Our method can be applied to any *raw* Hi-C data to analyze the multi-scale domain organization.

## Results

Our primary contribution is a method for characterizing the chromatin domains (CDs) at multiple scale in a unified framework.

### Correlation-based model for chromatin domains (CDs)

Our approach is based on an assumption that the chromosome is organized into discrete *chromatin domains* (CDs), such that the position (and consequently, the activity) of two genomic loci within the same CD is more strongly correlated than across different CDs.

The pattern of pairwise correlation between different genomic loci is captured by Hi-C. However, most standard analysis approaches do not explicitly address how the information in Hi-C should be interpreted in a statistically rigorous way. Here we start by constructing a bridge between the observed data (Hi-C counts) and the key statistical quantity in our model (the pairwise correlation matrix).

We use a Gaussian polymer network model to describe the long-range pairwise interaction of genomic segments in a chromosome. Use of the Gaussian polymer network model was motivated by an observation, from fluorescence measurements, that the spatial distances between pairs of chromatin segments are well described by the gaussian distribution [21, 42–44] (see S1 Fig). This observation suggests that, despite the presence of cell-to-cell variation in a population of cells [39, 45–51], we can still approximate the chromosome with a gaussian polymer network whose configuration fluctuates around a local basin of mechanical equilibrium [16, 41, 52–55], for the purpose of modeling the pairwise distances. See S1 Appendix for more discussion.

The Gaussian polymer network provides a flexible physical model for diverse patterns of correlation, because we allow each pair of segments to interact with a harmonic potential with an independent "spring constant"—the matrix of these spring constants can be translated to the pairwise correlation matrix between the segments. At the same time, the model is tractable enough to allow us to write down the pairwise distance distribution given the pairwise correlation, and consequently the pairwise contact probability. Assuming that Hi-C is essentially a manifestation of the pairwise contact probabilities, we can construct a chain of relationships that connects the Hi-C data matrix to the pairwise correlation matrix (Fig 2A). See Methods for details.

Now we have a pairwise correlation matrix, $\mathbf{C}$, that summarizes the experimental data. We assume that the correlation pattern in $\mathbf{C}$ was *generated* from a hidden model, which is characterized by a relatively small number of chromatin domains (CDs). Suppose that there are $N$ genomic segments, and that each segment $i \in \{1, 2, \cdots, N\}$ belongs to one of $K$ domains, indexed by $s_i \in \{1, 2, \cdots, K\}$. The vector $\mathbf{s} = (s_1, s_2, \ldots, s_N)$ can be called the *domain solution* in this model, because it summarizes how the genome is organized into distinct CDs. We are assuming that pairs of segments in the same CD will be more highly correlated (corresponding elements in $\mathbf{C}$ have larger values), which are manifested in higher counts in the Hi-C data. Adapting the *group model* [56–58] from statistical mechanics, we formulate the pairwise correlation between two genomic segments $(i, j)$ to have two contributions: the intra-domain correlation of the $s_i$-th CD that is present only when $s_i = s_j$, and the domain-independent correlation between the two segments (see Methods).

The goal of our method is to identify the domain solution $\mathbf{s}$ that best represents the pattern in the correlation matrix $\mathbf{C}$. To evaluate how well a domain solution $\mathbf{s}$ explains the correlation pattern in the data, we calculate the log likelihood $\mathcal{L}(\mathbf{C}; \mathbf{s})$ that the observed correlation $\mathbf{C}$ was drawn from an underlying set of domains $\mathbf{s}$. We also impose an additional preference to more parsimonious solutions; instead of explicitly fixing the number of domains, $K$, we penalize solutions with more fragmented domains, in terms of the *generalized* number of domains $\mathcal{K}(\mathbf{s})$. We combine the two measures into a single family of objective functions, parameterized by $\lambda$, and look for solution $\hat{\mathbf{s}}$ that maximizes:

$$\hat{\mathbf{s}}_\lambda = \max_{\mathbf{s}}[\mathcal{L}(\mathbf{C}; \mathbf{s}) - \lambda\mathcal{K}(\mathbf{s})] \tag{1}$$

(see Methods for the mathematical details and the optimization procedure). The free parameter $\lambda$ ($\geq 0$) controls how strongly the problem prefers parsimonious solutions: with a larger $\lambda$, the solution $\hat{\mathbf{s}}$ for the optimization problem would tend to have fewer numbers of domains. When $N$ is fixed, we can also say that a larger $\lambda$ prefers solutions with domains that are larger

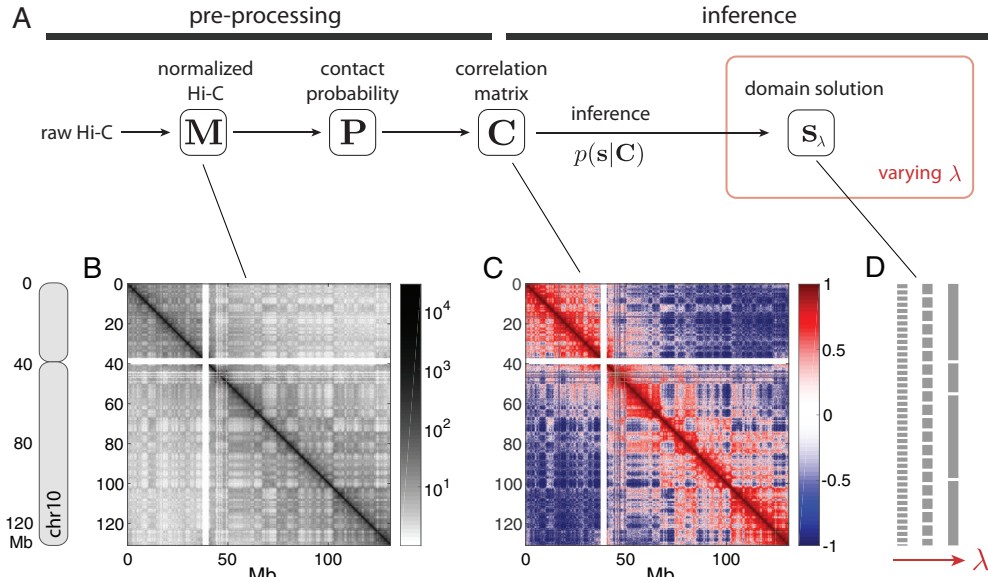

**Fig 2. An overview of the Multi-CD method.** (**A**) We first pre-process Hi-C data to extract a correlation matrix **C**. Given **C**, we infer the chromatin domain (CD) solutions **s** at multiple scales by varying a single parameter λ. At each λ, the best domain solution is found through simulated annealing, in which the effective temperature $T$ is gradually decreased (inner blue box). A complete Multi-CD algorithm involves repeating the process for different values of λ (outer red box), to obtain a family of solutions. (**B**) An example of the normalized Hi-C matrix, (**C**) and the correlation matrix that results from pre-processing. Shown is the full chromosome 10 in GM12878 (50-kb Hi-C), which is used as an example dataset throughout the paper. (**D**) Simplified schematic for the resulting family of domain solutions, $\{\mathbf{s}_\lambda\}$, at varying parameter λ. Each **s** is a vector of domain indices; line breaks illustrate domain boundaries. These solutions are not meant to be the optimal solutions for the shown data, but they illustrate how the typical domain scale increases with λ.

in the genomic scale. Solving at multiple values of λ (Fig 2B), therefore, would reveal the *multi-scale* domain structure in data.

There is a mathematical analogy between the form of the objective function in Eq 1, and the grand canonical ensemble in statistical physics (see Methods for details). The analogy to the well-studied physical formulation provides a useful conceptual framework, and justifies the use of efficient inference procedures such as simulated annealing (see Methods). We also note that, in this view, the parameter λ has a physical interpretation as the effective *chemical potential* of a domain in the solution, which is associated with the creation of a new domain or the merging of two domains into one.

Putting together, we present Multi-CD, a unified framework for modeling, inferring and interpreting the hidden chromatin domains (CDs) from Hi-C data.

## Discovery of CDs at multiple scales

Now we show how our method can be used to characterize the multi-scale structure of the chromosome and generate new insights. We applied Multi-CD to a sample subset of Hi-C data from a commonly used human lymphocyte cell line, GM12878, at 50-kb resolution. After the transformation of the raw Hi-C (Fig 3A) into a correlation matrix (Fig 3B), Multi-CD was employed to infer a family of CD solutions that vary with λ (Fig 3C). We also applied Multi-CD to four other cell lines, HUVEC, NHEK, K562, and KBM7 (Fig 3D), and analyzed the CD solutions for all five cell lines. All results shown in the main text consider chromosome 10. See See S2 Fig for similar results from three other chromosomes (chr4, 11, 19).

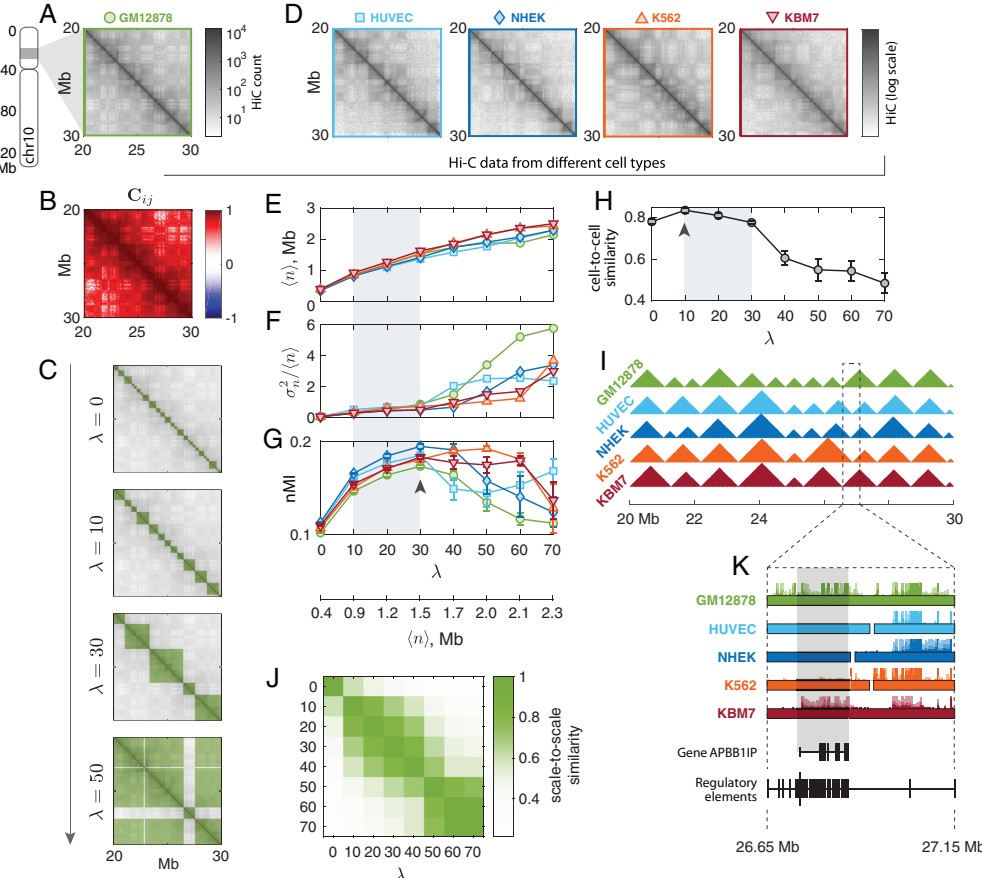

**Fig 3. Multi-scale chromatin domain solutions for various cell types.** (**A**) A subset of 50-kb resolution Hi-C data, covering a 10-Mb genomic region of chr10 in GM12878. (**B**) The cross-correlation matrix $\mathbf{C}_{ij}$ for the corresponding subset. (**C**) Multi-CD applied to the correlation matrix in **B**. Domain solutions determined at 4 different values of $\lambda$ = 0; 10; 30; 50. (**D**) Hi-C data from the same chromosome (chr10) in four other cell lines: HUVEC, NHEK, K562, and KBM7. Same subset as in **A**. (**E-G**) Characteristics of the domain solutions determined for all five cell lines in **A** and **D**: (**E**) the average domain size, $\langle n \rangle$ (**F**) the index of dispersion in the domain size, $D(=\sigma_n^2/\langle n \rangle)$ (**G**) the normalized mutual information, nMI. (**H-I**) Comparison of domain solutions across cell types. (**H**) Average cell-to-cell similarity of domain solutions, in terms of Pearson correlations, at varying $\lambda$. (**I**) Domain solutions obtained at $\lambda$ = 10 for 5 different cell types. See S3 Fig for solutions at $\lambda$ = 0 and $\lambda$ = 40. (**J**) Similarity between domain solutions at different $\lambda$'s, shown for GM12878. See S4 Fig for corresponding results for the other four cell lines. (**K**) RNA-seq signals from the five cell lines (colored hairy lines), on top of the TAD solutions (filled boxes), in a genomic interval that contains the regulatory elements associated with a gene APBB1IP. APBB1IP is transcriptionally active only in two cell lines, GM12878 and KBM7, where the regulatory elements are fully enclosed in the same TAD. See S5 Fig for additional examples.

We observed several general features from the families of CD solutions in these cell lines:

(i) The average domain size $\langle n \rangle$ always increased monotonically with $\lambda$ (Fig 3E), as expected from our construction of the prior.

(ii) The domain sizes were relatively homogeneous in the small-domain regime (small $\lambda$), but became heterogeneous after a cross-over point (Fig 3F). To quantify this, we defined the *index of dispersion* for the domain sizes, i.e., the variance-to-mean ratio $D = \sigma_n^2/\langle n \rangle$. If the domains were generated by randomly selecting the boundaries along the genome, $D = 1$; a smaller $D < 1$ indicates that domain sizes are more homogeneous than random. A larger $D > 1$ means that the domain sizes are heterogeneous. The crossover points arise at

$\lambda_{cr} \approx 30 - 40$ ($\langle n \rangle_{cr} \approx 1.6$ Mb) for GM12878, HUVEC, and NHEK; and $\lambda_{cr} \approx 60 - 70$ ($\langle n \rangle_{cr} \approx 2.2$ Mb) for K562 and KBM7. We observed that the onset of heterogeneity was related to the appearance of non-local domains (Fig 3C).

(iii) We quantified the goodness of each CD solution by comparing its corresponding binary matrix against the Hi-C data in terms of the normalized mutual information (nMI; see Methods). There is a scale, $\lambda^*$, at which the diagonal block pattern manifested in Hi-C data is most accurately captured. In Fig 3G, the best solution was found at $\lambda^* \approx 30$ for GM12878, HUVEC, and NHEK; the $\lambda^*$'s were identified at larger values for K562 and KBM7. As an interesting side note, K562 and KBM7 belong to immortalized leukemia cell lines, whereas the other three cell types are normal cells; the different statistical property of Hi-C patterns manifested in $\lambda^*$ may hint at a link between the pathological state and a coarser organization of the chromosome.

(iv) The CD solutions inferred by Multi-CD, especially the families of local CDs, appeared to be *conserved* across different cell types (Fig 3H and 3I). We quantified the extent of domain conservation in terms of the Pearson correlation (Methods), averaged over all pairs of different cell types. Domain conservation was strong for smaller domains at $\lambda \leq 30$ ($\langle n \rangle \lesssim 1.5$ Mb), with the strongest conservation at $\lambda = 10$ (Fig 3H). The CDs at $\lambda = 10$ are shown in Fig 3I for five different cell types.

(v) Finally, we quantified the similarity between pairs of CD solutions obtained at different scales, again using the similarity measure based on Pearson correlation. In the case of GM12878, the family of CD solutions is divided into two regimes; the smaller-scale CD solutions from a range of $10 \leq \lambda \leq 40$ are correlated among themselves, and the larger-scale CD solutions from $\lambda > 40$ as well. CD solutions below and above $\lambda \approx 40$ are not correlated with each other (Fig 3J; also see S4 Fig for the other cell lines).

The division boundary in (v) is found at a $\lambda$ value in the similar range with the best-clustering scale $\lambda^*$, and the crossover $\lambda_{cr}$ from local/homogeneous to non-local/heterogeneous CDs (compare S4 Fig to Fig 3F and 3G). Hereafter we will refer to the two regimes as the family of *local* CDs with homogeneous size distribution ($\lambda \lesssim \lambda^*$), and the family of *non-local* CDs with heterogenous size distribution ($\lambda \gtrsim \lambda^*$).

## TAD-like organizations in the family of local CDs

We identified at least three important scales in the family of local CDs. First of all, there is a scale at which domain conservation is maximized across different cell types ($\lambda = 10$). This observation is consistent with the widely accepted notion that TADs are the most well-conserved, common organizational and functional unit of chromosomes, across different cell types [27, 59]. Thus, for the example from human chromosome 10, we identify the CDs found at this scale $\lambda = 10$ as the TADs. The average domain size at $\lambda = 10$ is $\langle n \rangle \approx 0.9$ Mb, which agrees with the typical size of TADs as suggested by previous studies [22, 23].

The goodness of clustering, on the other hand, is maximized at a larger scale, $\lambda \approx 30$ ($\langle n \rangle \approx 1.5$ Mb) for GM12878. The CDs at this scale turn out to be aggregates of multiple TADs in the genomic neighborhood, from visual inspection (see Fig 3C), or as quantified in terms of a nestedness score (Methods). We therefore identify these CDs as the "meta-TADs", a higher-order structure of TADs, adopting the term of Ref. [8]. In contrast to a previous analysis that extended the range of meta-TADs to the entire chromosome [8], we use the term meta-TAD exclusively for the larger-scale local CDs, distinguishing them from the non-local structures (i.e., compartments, discussed below). We note, however, that the terminologies of TADs and

the meta-TADs are still not definitive—a recently proposed algorithm based on structural entropy minimization [60] found that the "best" solutions were found at $\sim$ 2 Mb domains, which is consistent with our findings, although these domains were called the TADs in Ref. [60].

Finally, a trivial but special scale is $\lambda = 0$, where no additional preference for coarser CDs is imposed. The CDs at this scale are supposed to best explain the local correlation pattern that is reflected in the strong Hi-C signals near the diagonal. These smaller CDs are almost completely nested in the TADs and the meta-TADs; we can therefore call them the sub-TADs. We also confirm that the sub-TAD solutions are not limited by the resolution of the Hi-C data; sub-TADs are robustly reproduced from a finer, 5-kb Hi-C (S6 Fig).

The first three panels in Fig 3C shows three representative TAD-like CD solutions at $\lambda \leq \lambda^*$: sub-TADs ($\lambda = 0$; smallest CDs), TADs ($\lambda = 10$, strongest domain conservation), and meta-TADs ($\lambda = \lambda^* = 30$, largest nMI). The nested structure is reminiscent of the hierarchically crumpled structure of chromatin chains [37].

## Chromatin organization and its link to gene expression

The CD solutions from Multi-CD can provide important insights into the link between chromatin organization and gene expression. To demonstrate this, we overlaid the RNA-seq profiles on the TAD solutions, identified for the corresponding subset of the chr10 of five cell lines (GM12878, HUVEC, NHEK, K562, KBM7) (Fig 3K). At around 26.8 Mb position of this chromosome, we found a gene APBB1IP, which is transcriptionally active in GM12878 and KBM7 but not in HUVEC, NHEK and K562. Consulting the GeneHancer database [61], we identified the regulatory elements for this gene (enhancers and promoters) in the interval between 26.65 and 27.15 Mb. Notably, our Multi-CD solutions show that for the case of APBB1IP gene the interval associated with the regulatory elements is fully enclosed in the same TAD in GM12878 and KBM7, whereas it is split into different TADs in the other three cell lines (Fig 3K). This is an important demonstration of how the 3D chromosome structure regulates the gene expression level, which is also consistent with the understanding that TADs are the functional units of the genome [7, 8, 17, 20]. More generally, the observation suggests that gene expression depends on the spatial organization of the genome, captured by our TAD solutions, which constrains the interaction between the gene and the regulatory elements (also see S5 Fig for additional examples of TAD structure-dependent cell type specific gene expression).

## Compartments as the best domain solution that coexists with TAD-like domains

Looking at the correlation pattern in Hi-C data (for example Fig 2B and 2C), one can hardly miss the prominent global structure with a characteristic checkerboard pattern demonstrated in the off-diagonal part of the matrix. These highly non-local, super-Mb-sized domains are generally defined as the compartments in the chromosome organization [27]. Given the large scale of the compartments, compartment-like solutions were initially anticipated in the large-$\lambda$ limit of Multi-CD. However, a naïve application of Multi-CD by increasing $\lambda$ failed in identifying the compartments; some non-local CDs were found (Fig 3C), but the characteristic pattern of compartments was not obtained by merely increasing the value of $\lambda$.

We hypothesized that compartments correspond to a *secondary* CD solution that coexists with the primary solution (in this case the TAD-like domains, as already identified), rather than belonging to the same family of solutions. As described in Methods, such secondary solution can be inferred by applying an extended version of Multi-CD, effectively taking out the contribution of the primary solution from the correlation matrix **C**. We consider a simplified

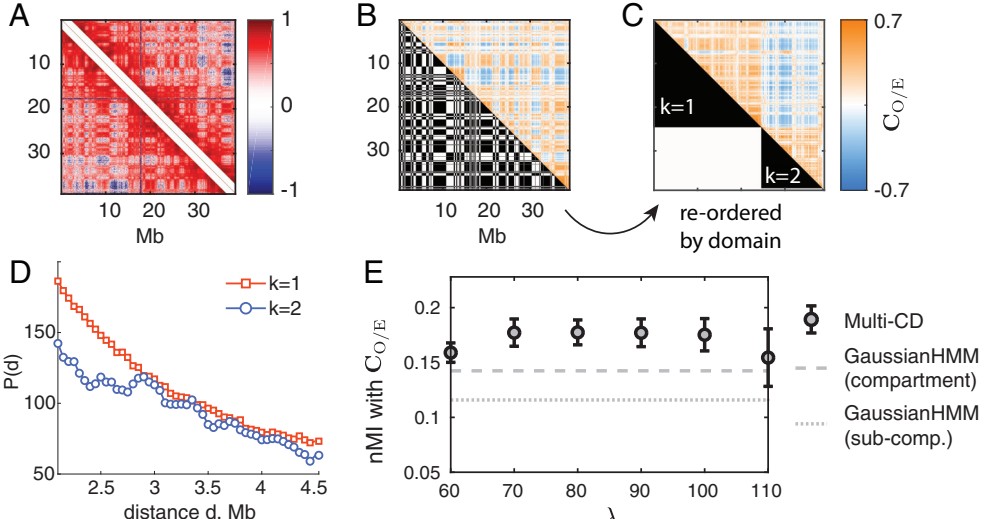

**Fig 4. Domain solutions for compartments.** (**A**) Input correlation data for compartment identification. The 2-Mb diagonal band was removed. (**B**) Lower triangle: CDs obtained at $\lambda = 90$, based on the diagonal-band-removed data, which we identify as the compartments. Upper triangle: the $\mathbf{C}_{O/E}$ matrix shown for comparison. (**C**) Same pair of data, after re-ordering to collect the two largest CDs in our solution, $k = 1$ and $k = 2$ (lower triangle). The $\mathbf{C}_{O/E}$ is simultaneously reordered to show a clear separation of correlation patterns (upper triangle). (**D**) Intra-domain contact profiles for the two CDs $k = 1$ and $k = 2$. The domain solution $k = 1$ is locally more compact, with more contacts at short genomic distances. We therefore identify $k = 1$ as the B-compartment, and $k = 2$ as the A-compartment. (**E**) nMI between CDs at varying $\lambda$ and $\mathbf{C}_{O/E}$, showing a plateau in the range $70 \leq \lambda \leq 100$. CDs inferred by Multi-CD show consistently higher nMI, compared to sub-compartments (dashed line) and compartments (dotted line) from a previous method [19].

version of this problem, and remove from **C** a diagonal band of width 2 Mb, corresponding to the known size of meta-TADs (Fig 4A). Along with our prior knowledge of the primary solution, this approximation has the advantage that the secondary inference (for compartments) can be performed independently from the outcome of the primary inference (for the TAD-like domains). In this case, the algorithm successfully captures the non-local correlations, and identifies two large compartments with alternating patterns (Fig 4B). The correspondence is clearer when the indices of segments are re-ordered (Fig 4C). Because the larger CD ($k = 1$) shows a greater number of contacts (Fig 4D), it can be associated with the B-compartment, which is usually more compact; $k = 2$ is associated with the A-compartment. Further validation of the two compartments will be presented below, through comparisons with epigenetic markers.

To compare the goodness of our compartments with existing methods, we calculated the nMI against the $\mathbf{C}_{O/E}$ matrix, the conventional form for compartment identification (Methods) [18]. We find that Multi-CD outperforms GaussianHMM [19], a widely accepted benchmark algorithm, in capturing the large-scale structures in Hi-C (Fig 4E).

## Multi-scale, hierarchical organization of CDs

Now that we identified four classes of CD solutions, namely sub-TADs, TADs, meta-TADs and compartments, we examined their relationships. Note that these CDs were obtained independently at the respective $\lambda$ values, not through a hierarchical merging. Sub-TADs or TADs are almost always nested inside a meta-TAD, and TADs inside a meta-TAD, whereas there are mismatches between the TAD-like domains and the compartments (Fig 5). We quantified this relationship in terms of a nestedness score $h$, such that $h = 0$ indicates the chance level and

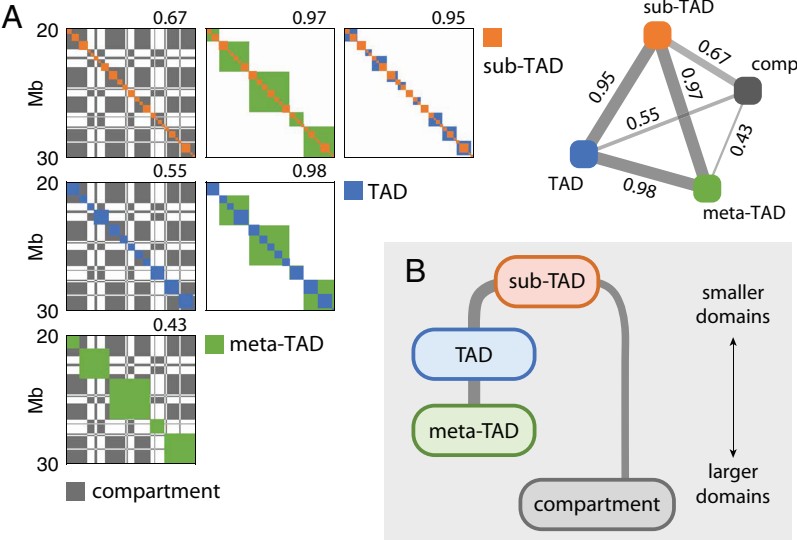

**Fig 5. Hierarchical organization of CD families.** (**A**) Hierarchical structure of CDs are highlighted with the domain solutions for sub-TADs (red), TADs (green), meta-TADs (blue) and compartments (black). Shown for chr10 of GM12878. Each square panel overlays a pair of CD solutions; number above the panel reports the nestedness score. Inset: a reprint of the nestedness scores in a tetrahedral visualization with the four representative CD solutions. (**B**) A schematic diagram of inferred hierarchical relations between sub-TADs, TADs, meta-TADs and compartments, based on our calculation of nestedness scores.

$h = 1$ a perfect nestedness (Methods), along with a visual comparison of each pair of CD solutions (Fig 5A). This analysis confirms that there exists an appreciable amount of hierarchy between any pair of TAD-like domains (sub-TADs, TADs, and meta-TADs). On the other hand, the hierarchical links between the TAD-like domains and compartments are much weaker, which is again consistent with the recent reports that TADs and compartments are organized by different mechanisms [62, 63].

Although the nestedness score between sub-TADs and compartments (nestedness score $h = 0.67$) is not so large as those among the pairs of TAD-based domains, it is still greater than those between TADs and compartments ($h = 0.55$) or between meta-TADs and compartments ($h = 0.43$). Thus, sub-TAD can be considered a common building block of the chromatin architecture (see Fig 5B).

## Validation of domain solutions

The CD solutions from Multi-CD are in good agreement with the results of several existing methods that specialize in particular domain scales. Specifically, our CDs correspond to the previously proposed sub-TADs [19] at $\lambda = 0$, to the TADs [22] at $\lambda \approx 10$, and to the compartments [19] at $\lambda \approx 90$ (S7 Fig). When assessed in terms of the nMI, Multi-CD outperforms the corresponding alternatives (ArrowHead [19], DomainCaller [22], GaussianHMM [19] for sub-TADs, TADs, meta-TADs) at the respective scales (Fig 6A).

We also compared our CD solutions with several biomarkers that are known to be correlated with the spatial organization of the genome [64]. All results shown here are for chr10 of GM12878.

First, we calculated how much the boundaries of our sub-TAD and TAD solutions are correlated with the CTCF signals, which are known to be linked to TAD boundaries [22, 23] (Fig 6B). We quantified this in terms of a correlation function, $\chi(d)$, where $d$ is the genomic

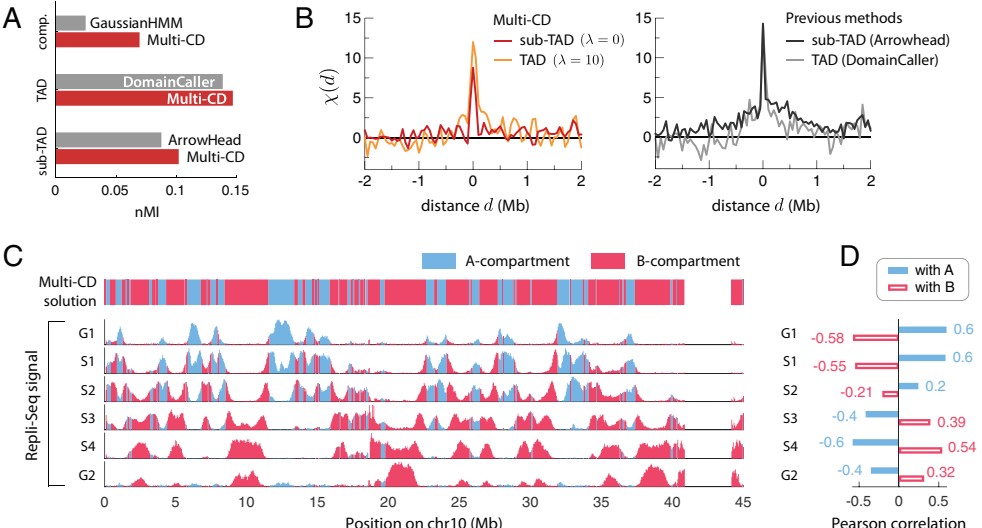

**Fig 6. Validation of CD solutions from Multi-CD.** (**A**) In terms of the normalized mutual information between the CD solutions and the input data, Multi-CD outperforms ArrowHead, DomainCaller and GaussianHMM at the corresponding scales (sub-TAD, TAD and compartment). (**B**) The correlation function $\chi(d)$ between CTCF signals and the domain boundaries. Shown for sub-TADs and TADs, obtained from Multi-CD (left); from ArrowHead and DomainCaller (right). (**C**) Genome-wide, locus-dependent replication signal. Top panel shows the A- (blue) and B- (red) compartments inferred by Multi-CD. Bottom panels show the replication signals in six different phases in the cell cycle, shaded in matching colors for the two compartments. (**D**) Pearson correlation between the replication signals and the two compartments A (filled blue) and B (open red).

distance between a domain boundary and each CTCF signal (Methods). Compared with those of ArrowHead and DomainCaller, the correlation function calculated for the results from Multi-CD shows a similar enrichment of CTCF signals at domain boundaries (high peak of correlation at $d \approx 0$), along with better precisions (fast decay of correlation as $d$ increases) (see Fig 6B). Specifically, when fitted to exponential decays, the correlation lengths are 34 kb ($\lambda = 0$) and 143 kb ($\lambda = 10$) for Multi-CD, compared to $\gtrsim 900$ kb for the two previous methods (Fig 6B).

Next, we compared our compartment solutions (CDs at $\lambda = 90$, shown in Fig 4B) with the replication timing profiles (Repli-Seq), which are known to correlate differently with the A- and B- compartments [9, 65]. Our inferred compartments exhibit the anticipated patterns of replication timing (Fig 6C); the A-compartment shows an activation of replication signals in the early-phases (G1, S1, S2) and a repression in the later phases (S3, S4, G2), whereas the B-compartment shows an opposite trend. There is a clear anti-correlation between the replication patterns in the two compartments along the replication cycle (Fig 6C), which is also quantified in terms of the Pearson correlation (Fig 6D). Comparison to other epigenetic markers, such as the pattern of histone modifications, further confirms the association of our CD solutions with the A/B-compartments (S8 Fig).

## Discussion

Multi-CD is a unified framework for Hi-C data analysis and a principled interpretation of Hi-C from the viewpoint of polymer and statistical physics, enabling identification of CDs at various genomic scales. As a computational algorithm, Multi-CD includes two core steps: the pre-processing of raw Hi-C data into a correlation matrix, and the inference of chromatin domain (CD) solutions from the correlation matrix.

The pre-processing, based on a model of the chromosome as a gaussian polymer network, allows us to make a physically justifiable interpretation of the Hi-C data. This fundamentally differentiates our approach from other methods in which the target patterns need to be selected empirically. Moreover, the polymer physics-based transformation obviates the need for the arbitrary use of a nonlinear (most often logarithmic) scaling to enhance the correlation patterns in the Hi-C data.

The inference problem is rigorously formulated based on a statistical mechanical model of the pairwise correlation matrix in the presence of hidden correlated domains. We note that in the current model, domains are defined very generally in terms of a many-to-one grouping of the chromatin segments. As a result, our method deals with *non-local* CDs (occupying non-consecutive locations on the genome) as naturally as it does with the *local* CDs (consecutive intervals between two boundaries on the genome). This differentiates Multi-CD from previous methods that focus on local features in Hi-C, such as CD boundaries or loops enriched with higher contact frequencies.

As an important feature, Multi-CD offers a one-method-fits-all framework to identify CDs at multiple scales, tuning a single parameter $\lambda$ to control the preference to coarser solutions. Because the resulting family of CD solutions share the same formulation, it is possible to make quantitative comparisons between CD solutions at different scales. Moreover, Multi-CD can find CDs across a wide range of scales without having to adjust or down-sample the Hi-C data to match the scale of CDs to be identified, which is an important improvement over many existing methods.

Applying Multi-CD to Hi-C data from five human cell lines, our analysis revealed special scales at which the CD solutions are particularly interesting: sub-TADs ($\lambda = 0$), TADs ($\lambda = 10$, where domain conservation was strongest), and meta-TADs ($\lambda = 30$, where the correlation pattern was best captured). At larger scales, we found that compartments ($\lambda = 90$) emerge as a secondary solution that can be inferred after explaining away the contributions from the TAD-like solutions. We confirmed that Multi-CD successfully reproduces, or even outperforms, the existing methods to identify CDs at the specific scales. Importantly, Multi-CD achieves this performance through a single unified algorithm, which not only identifies the specific CD solutions accurately, but also allows a comparative analysis of the multi-scale family of solutions.

In particular, we characterized the hierarchical organization of the chromatin by quantifying the similarity and the nestedness between CD solutions at two different scales. We showed that the characteristics of CD solutions shared by the local, TAD-like domains do not precisely hold together in the non-local, compartment-like domains. This finding is consistent with the recent studies which report that compartments and TADs are formed by different mechanisms of motor-driven active loop extrusion and microphase separation, and that they do not necessarily have a hierarchical relationship [63, 66–68]. It is also consistent with our modeling assumptions that compartments are the secondary solution, whereas the TAD-like domains belong to the family of primary solutions. The ability of our model to appropriately describe the secondary solution, in addition to a family of primary solutions, further highlights the strength of our framework. Meanwhile, the sub-TADs are nested in each of the other three solutions, including the compartments (Fig 5), indicating that sub-TADs are the fundamental building blocks of the higher-order CD organization. We note that this is not a trivial consequence of the finite data resolution; the sub-TADs are robustly recovered when Multi-CD was applied to Hi-C data at a 10-fold finer resolution of 5kb (S6 Fig).

While there are methods that report hierarchical CDs [33, 34], Multi-CD makes significant advances both algorithmically and conceptually. Multi-CD can detect non-local domains with better flexibility instead of finding a set of intervals. Multi-CD also avoids the high

false-negative rate that is typical of the previous method (e.g., TADtree [33]) that focuses on the nested domain structure (S9 Fig). Further, employing an appropriate prior to explore the solution space effectively, Multi-CD can avoid the problem encountered in Armatus [34] which skips detection of domains in some part of Hi-C data while its single scale parameter is varied (S9 Fig).

Multi-CD is a method of great flexibility that can be readily applied to analyze any dataset that exhibits pairwise correlation patterns. However, two cautionary remarks are in place for more careful interpretation of the results. (i) In general, the relevant values of λ depend on the resolution of the input Hi-C dataset, as well as on the cell type. While λ is a useful parameter that allows comparative analysis, its specific value does not carry any biological significance. Although we referred to a specific CD solution by the corresponding value of λ in the current analysis (Fig 3), the lesson should *not* be that TADs, for instance, always correspond to the particular value of λ; instead, TADs should be identified as the most conserved CD solutions across cell types after scanning a range of λ's. (ii) Multi-CD is agnostic about whether the collected data is homogeneous or heterogeneous. Application of Multi-CD to single-cell Hi-C data, and the subsequent interpretation of the result, would be straightforward; however, if the input Hi-C data were an outcome of a mixture of heterogeneous subpopulations, the solution from Multi-CD would correspond to their superposition. This is a fundamental issue inherent to any Hi-C data analysis method. Nevertheless, the population-averaged pattern manifest in Hi-C carries a rich set of information that is specific to the cell type. The need for interpretable inference methods that can extract valuable insights into the spatial organization of the genome, including ours, is still high.

To recapitulate, in order to glean genome function from Hi-C data that varies with the genomic state [12–15], a computationally accurate method for characterizing the domain organization is of vital importance. Multi-CD is a physically principled method that identifies the multi-scale structure of CDs, by solving a family of optimization problems with a single tuning parameter. We find the resulting CD solutions in excellent match with biological data such as CTCF binding sites and replication timing signal. Our framework enables quantitative analyses of CD structures identified across multiple genomic scales and various cell types, offering general physical insights into the chromatin organization inside cell nuclei.

## Methods

### Interpretation of Hi-C data

We first describe our physical interpretation of Hi-C. We construct a polymer network model of the chromosome, and assume that Hi-C is essentially a sampling procedure for the contact probability between pairs of segments in the network.

**Gaussian polymer network model for the chromosome.**   For a polymer chain whose long-range pairwise interactions are restrained via harmonic potentials with varying stiffness, the probability density function of the distance between a pair of segments $i$ and $j$ is written in the following form:

$$P(r_{ij}; \gamma_{ij}) = \frac{4}{\sqrt{\pi}} \gamma_{ij}^{3/2} r_{ij}^2 \exp(-\gamma_{ij} r_{ij}^2), \tag{2}$$

where $\gamma_{ij}$ amounts to the "stiffness" or the "spring constant" of the harmonic potential, up to a factor of the energy unit $k_B T$. (Note that this $T$ is the physical temperature, which should not be confused with the effective temperature for the simulated annealing in the inference procedure). This $\gamma_{ij}$ is directly related to the positional covariance determined by the topology of polymer network [16, 69]. More specifically, we describe the spatial positions of the polymer

segments using a gaussian distribution with zero mean and and a covariance matrix $\Sigma$, with elements $(\Sigma)_{ij} = \sigma_{ij} = \langle \delta \mathbf{r}_i \cdot \delta \mathbf{r}_j \rangle$. Then it follows that the distance $r_{ij} = |\mathbf{r}_i - \mathbf{r}_j|$ between two different monomers $i$ and $j$ ($i \neq j$) can be described in the form of a weighted gaussian function (Eq 2) where the variance $(2\gamma_{ij})^{-1}$ is associated with the covariance matrix elements as

$$\gamma_{ij}^{-1} = 2(\sigma_{ii} + \sigma_{jj} - 2\sigma_{ij}). \tag{3}$$

The contact probability between the two segments, $p_{ij}$, is the probability that their pairwise distance $r_{ij}$ is below a cutoff distance $r_c$. The contact probability is then calculated as $p_{ij} = \int_0^{r_c} dr\, P(r; \gamma_{ij})$. Once we obtain the contact probability $p_{ij}$ from the Hi-C data (more details below), this model allows us to reversely solve for the correlation matrix $\mathbf{C}$. Specifically, there is a one-to-one mapping from the contact probability $p_{ij}$ to the stiffness parameter $\gamma_{ij}$, which allows one to determine the covariance matrix $\{\sigma_{ij}\}$, and consequently the cross-correlation matrix, $(\mathbf{C})_{ij} = \sigma_{ij}/\sqrt{\sigma_{ii}\sigma_{jj}}$ with $-1 \leq (\mathbf{C})_{ij} \leq 1$ for $i \neq j$ and $(\mathbf{C})_{ii} = 1$ (correlation of the self-interaction is unity) (more details below). In summary, the following transformation from $p_{ij}$ to $(\mathbf{C})_{ij}$ is conceived (see also Fig 2):

$$p_{ij} \longrightarrow \gamma_{ij} \longrightarrow \sigma_{ij} \longrightarrow (\mathbf{C})_{ij}. \tag{4}$$

**Normalization and contact probability.** Here we describe how the Hi-C data can be interpreted as a set of contact probabilities for pairs of genomic segments, $p_{ij}$. Typically, a Hi-C matrix have widely varying row-sums; for example, the net count of the $i$-th segment in the experiment is much larger than the net count of the $j$-th segment. To marginalize out this site-wise variation and only focus on the differential strengths of pairwise interactions, the raw Hi-C matrix $\mathbf{M}_{\text{raw}}$ is normalized to have uniform row and column sums. This is achieved using the Knight-Ruiz (KR) algorithm [70], which finds a vector $\mathbf{v} = (v_1, \cdots, v_N)$ for calculating $(\mathbf{M}_{ij} = v_i v_j (\mathbf{M}_{\text{raw}})_{ij}$, such that each row (column) in $\mathbf{M}$ sums to 1.

We assume that the normalized Hi-C signal is proportional to the contact probability: $(\mathbf{M})_{ij} \propto p_{ij}$. Note that $p_{ij}$ is the probability that the two segments $i$ and $j$ are within a contact distance, and the rows of the contact probability matrix $(\mathbf{P})_{ij} = p_{ij}$ is not required to sum to 1. Because the proportionality constant is unknown *a priori*, however, we have a free parameter to choose. We do this by fixing the average nearest-neighbor contact probability, $\bar{p}_1 = \langle p_{i,i+1} \rangle$. We expect the $\bar{p}_1$ to be relatively close to 1, assuming that nearest-neighbor contact is likely; but not exactly 1, because there are variations among the nearest-neighbor Hi-C signal. In this work we chose $\bar{p}_1 = 0.9$. The resulting contact probability matrix $\mathbf{P}$ is given as $(\mathbf{P})_{ij} = \min(1, \hat{p}_{ij})$, with $\hat{p}_{ij} = (\bar{p}_1/\mu)(\mathbf{M})_{ij}$, where $\mu = \langle \mathbf{M}_{i,i+1} \rangle$ is the Hi-C signals averaged over the nearest-neighbors. At $\bar{p}_1 = 0.9$, in our case, the fraction of over-saturated elements ($\hat{p}_{ij} > 1$) was sufficiently small.

**Building the correlation matrix from Hi-C.** Here we continue to build the connection between the contact probability and the correlation matrices. The contact probabilities can be calculated from the distribution of pairwise distances (Eq 2), by saying that two segments $i$ and $j$ are in contact when their distance $r_{ij}$ is below a cutoff, $r_c$. In other words, we write

$$p_{ij} = \int_0^{r_c} P(r_{ij}; \gamma_{ij}) dr_{ij} = \mathrm{erf}(\gamma_{ij}^{1/2} r_c) - 2r_c\sqrt{\frac{\gamma_{ij}}{\pi}} e^{-\gamma_{ij} r_c^2}, \tag{5}$$

where $\mathrm{erf}(x) = \frac{2}{\sqrt{\pi}} \int_0^x dt\, e^{-t^2}$. Because this $p_{ij}$ is a monotonically increasing function of $\gamma_{ij}$, the value of $\gamma_{ij}$ is uniquely determined for each $p_{ij}$. It is straightforward to perform this inverse

mapping numerically; see our publicly available code for a simple Matlab implementation of the procedure.

Once we have the $\gamma_{ij}$'s, we can reconstruct the covariance matrix $\{\sigma_{ij}\}$ using Eq 3. Depending on the Hi-C pipeline, the Hi-C-derived contact matrix, $p_{ij}$, may or may not contain the diagonal entries, which reflect the contact frequency within each genomic region; however, regardless of the value of $p_{ii}$, the correlation of self-interaction is unity by definition $((\mathbf{C})_{ii} = 1)$. Note that although the value of $\gamma_{ij}$ depends on the choice of $r_c$, its effect is only to scale the $\gamma_{ij}$'s as $\gamma_{ij} \rightarrow r_c^2 \gamma_{ij}$, and consequently the $\sigma_{ij}$'s.

Finally, we normalize the covariance matrix to build the correlation matrix $\mathbf{C}$:

$$(\mathbf{C})_{ij} = \frac{\sigma_{ij}}{\sqrt{\sigma_{ii}\sigma_{jj}}} = \frac{\sigma_{ij}}{\sigma_c} = 1 - \frac{1}{4\sigma_c\gamma_{ij}}. \tag{6}$$

where we assume a uniform variance $\sigma_{ii} = \sigma_{jj} = \sigma_c$ along the diagonal, so as to set the overall intensity of $\mathbf{C}$. Here, we chose the value of $\sigma_c$ as the median of $1/4\gamma_{ij}$, i.e., $\sigma_c = \text{median}(1/4\gamma_{ij})$. This choice of $\sigma_c$ was motivated to ensure that the resulting $\mathbf{C}$ has balanced fractions of positive and negative correlation, making the most of the value range $[-1, 1]$ such that the global correlation pattern is clearly visible. This also cancels out the scaling effect of $r_c$ in $\sigma_{ij}$, so that the choice of $r_c$ does not affect the ultimate construction of the correlation matrix $\mathbf{C}$.

## Generative model for the correlation matrix

We now consider a generative model for the cross-correlation matrix $\mathbf{C}$ that displays correlated domains. Specifically, we adapt a statistical mechanical formalism known as the *group model* [56–58], which is used to cluster correlated domains in a given correlation data.

**The group model.** Let us assume that each genomic segment $i \in \{1, 2, \cdots, N\}$ belongs to a chromatin domain $s_i$. Then the vector $\mathbf{s} = (s_1, s_2, \ldots, s_N)$ can be called the *domain solution* for the $N$ segments. For example, a state $\mathbf{s} = (1, 1, 1, 2, 2, 3)$ describes a structure where the 6 genomic segments are clustered into 3 domains. Indexing of the domains is arbitrary. If there are $K$ distinct domains in the solution, we can always index the domains such that $s_i \in \{1, 2, \cdots, K\}$.

We also assume that the cross-correlation matrix $\mathbf{C}$ can be represented by the correlation between a set of hidden variables $\{x_i\}$ where $x_i$ denotes the "genomic state" of the $i$-th chromatin segment. Without loss of generality, we only consider the case where $x_i$ has zero mean and unit variance. Adapting the formalism in Refs. [56, 57], we assume that $x_i$ obeys the following stochastic equation

$$x_i = \sqrt{\frac{g_{s_i}}{1+g_{s_i}}}\eta_{s_i} + \frac{1}{\sqrt{1+g_{s_i}}}\epsilon_i \tag{7}$$

where $\eta_{s_i}$ and $\epsilon_i$ are two independent and identically distributed (i.i.d) random variables with $\eta_{s_i}, \epsilon_i \sim \mathcal{N}(0, 1)$, that are linked to the domain ($s_i$) and the individual segment ($i$) respectively. The clustering strength parameter $g_{s_i} (\geq 0)$ is associated with each domain $s_i$, such that a larger $g_{s_i}$ indicates a stronger contribution from the domain-dependent variable $\eta_{s_i}$. The cross-correlation between two segments $i$ and $j$ is written as

$$\langle x_i x_j \rangle = \frac{g_{s_i}}{1+g_{s_i}}\delta_{s_i s_j} + \frac{1}{1+g_{s_i}}\delta_{ij}. \tag{8}$$

In light of Eq 8, the first term of Eq 7 on the right hand side contributes to intra-CD correlation of the $s_i$-th CD; the second term of Eq 7 corresponds to the domain-independent noise. With a larger $g_{s_i}$, the domain $s_i$ becomes more clearly discernible from other domains ($s_j \neq s_i$).

We are interested in the inference problem in which the *model* correlation matrix $\langle x_i x_j \rangle$ is fitted against the *data* correlation matrix $\mathbf{C}$,

$$(\mathbf{C})_{ij} \Leftrightarrow \langle x_i x_j \rangle, \tag{9}$$

to find the best domain solution $s_i$ along with the corresponding clustering strength parameter $g_{s_i}$.

**Incorporating the secondary group to the group model.** The group model assumes that the correlation pattern in $\mathbf{C}$ is generated from a single level of group structure. However, there may be more complex situations in which the correlation pattern is hierarchical, not fully described by a single group. Here we expand the model to include up to the secondary group, whose contribution is, by construction, weaker than the primary group.

Suppose that the underlying grouping was bivariate, $i \mapsto (s_i, u_i)$, such that each genomic locus $i$ simultaneously belongs to a primary group $s_i$ and a secondary group $u_i$. Generalizing Eq 7, we assume a linear model

$$x_i = \frac{\epsilon_i + \sqrt{g_{s_i}}\, \eta_{s_i} + \sqrt{h_{u_i}}\, \xi_{u_i}}{\sqrt{1 + g_{s_i} + h_{u_i}}} \tag{10}$$

where $\xi_{u_i}$ and $h_{u_i}$ are, respectively, the random variable and the grouping strength parameter for the secondary group $u_i$. If we further assume that $s_i$ and $u_i$ are statistically independent, the pairwise correlation between two loci $i$ and $j$ can be written as

$$\langle x_i x_j \rangle = \frac{\delta_{ij} + g_{s_i} \delta_{s_i, s_j} + h_{u_i} \delta_{u_i, u_j}}{1 + g_{s_i} + h_{u_i}}; \tag{11}$$

the contributions from different groups are additive. We can rearrange Eq 11 to write

$$\frac{1 + g_{s_i} + h_{u_i}}{1 + h_{u_i}} \left( \langle x_i x_j \rangle - \frac{g_{s_i} \delta_{s_i, s_j}}{1 + g_{s_i} + h_{u_i}} \right) = \frac{\delta_{ij} + h_{u_i} \delta_{u_i, u_j}}{1 + h_{u_i}}, \tag{12}$$

such that the right-hand side becomes the single-group model. The left-hand side of this expression is a normalized residual of the correlation, which we will call $(\mathbf{C}^{\text{res}})_{ij}$. If we have already inferred the primary group $s_i$ and the corresponding strength $\tilde{g}_{s_i}$ without considering the secondary group, the correspondence to this two-group model (due to normalization) is given as $g_{s_i} = (1 + h_{u_i}) \tilde{g}_{s_i}$. Substituting this and replacing the model correlation $\langle x_i x_j \rangle$ with the data $(\mathbf{C})_{ij}$ simplify the left-hand side of Eq 12 to

$$(\mathbf{C}^{\text{res}})_{ij} = (1 + \tilde{g}_{s_i}) (\mathbf{C})_{ij} - \tilde{g}_{s_i} \delta_{s_i, s_j}. \tag{13}$$

Note that $\mathbf{C}^{\text{res}}$ is independent of the unknown secondary solution $u$ (and $h$). Now $u$ is the solution of a modified single-group problem, using this residual correlation $\mathbf{C}^{\text{res}}$ as the input data.

To infer the secondary group solution, therefore, one can simply repeat the inference procedure for the single-group problem after obtaining the primary group solution.

## Formulating the inference problem

Our goal is to identify the domain solution $\mathbf{s} = (s_1, s_2, \cdots, s_N)$ that best represents the pattern in the correlation matrix $\mathbf{C}$, with an appropriate set of strength parameters $\mathbf{g} = (g_1, g_2, \cdots, g_K)$, where $K$ is the number of distinct domains in the solution. Using the group model described above, we can calculate the likelihood $p(\mathbf{C}|\mathbf{s}, \mathbf{g})$ that the observed correlation matrix $\mathbf{C}$ was

drawn from an underlying set of domains $\mathbf{s}$ with strengths $\mathbf{g}$ [58] (See S2 Appendix for derivation). The log-likelihood, $\log p(\mathbf{C}|\mathbf{s}, \mathbf{g})$, is written as a sum over all domains in the solution $\mathbf{s}$:

$$\log p(\mathbf{C}|\mathbf{s}, \mathbf{g}) \quad = -\frac{1}{2}\sum_{k=1}^{K}\left[(1+g_k)\left(n_k - \frac{g_k c_k}{1+g_k n_k}\right) - n_k \log(1+g_k) + \log(1+g_k n_k)\right], \quad (14)$$

where $n_k = \sum_{i=1}^{N}\delta_{s_i,k}$ is the size of domain $k$, and $c_k = \sum_{i,j=1}^{N}\mathbf{C}_{ij}\delta_{s_i,k}\delta_{s_j,k}$ is the sum of all intra-domain correlation elements. The log-likelihood in Eq 14 is maximized at $\hat{g}_k = (c_k - n_k)/(n_k^2 - c_k)$ for each $k$, allowing us to consider the reduced likelihood $p(\mathbf{C}|\mathbf{s}) \equiv \max_{\mathbf{g}} p(\mathbf{C}|\mathbf{s}, \mathbf{g}) = p(\mathbf{C}|\mathbf{s}, \hat{\mathbf{g}})$.

For convenience, we write the likelihood function as $p(\mathbf{C}|\mathbf{s}) \propto \exp(-\mathcal{E}(\mathbf{s}|\mathbf{C})/T)$ to resemble a Boltzmann distribution, where $\mathcal{E}$ corresponds to the *energy* or to $-\mathcal{L}(\mathbf{C};\mathbf{s})$ introduced in Eq 1, and $T$ is the effective *temperature* that will be used later in the simulated annealing. At $\mathbf{g} = \hat{\mathbf{g}}$, $\mathcal{E}(\mathbf{s}|\mathbf{C})$ reads

$$\mathcal{E}(\mathbf{s}|\mathbf{C}) = \frac{1}{2}\sum_{k=1}^{K}\left[\log\frac{c_k}{n_k} + (n_k - 1)\log\frac{n_k^2 - c_k}{n_k^2 - n_k}\right]. \quad (15)$$

The problem of finding the maximum likelihood solution $\mathbf{s}$ is equivalent to finding the energy-minimizing $\mathbf{s}$ for $\mathcal{E}(\mathbf{s}|\mathbf{C})$.

Besides evaluating how well a domain solution $\mathbf{s}$ explains the correlation pattern in the data, we also want to impose an additional preference to more parsimonious solutions. Instead of explicitly fixing the number of domains, $K$, we modify the optimization problem using the method of Lagrange multiplier:

$$\min_{\mathbf{s}} \mathcal{H}_\lambda(\mathbf{s}|\mathbf{C}), \quad \mathcal{H}_\lambda(\mathbf{s}|\mathbf{C}) = \mathcal{E}(\mathbf{s}|\mathbf{C}) + \lambda\mathcal{K}(\mathbf{s}), \quad (16)$$

where $\mathcal{K}$ is a *generalized* number of domains. Specifically, we define $\mathcal{K}(\mathbf{s})$ as

$$\mathcal{K}(\mathbf{s}) = \exp\left(-\sum_{k=1}^{K} p_k \log p_k\right), \quad p_k = \frac{n_k}{N}, \quad (17)$$

such that $\log \mathcal{K}(\mathbf{s})$ is the entropy of $\mathbf{s}$. In particular, this quantity reduces to $\mathcal{K}(\mathbf{s}) = K$ in the regime where the domain sizes are uniform. The Lagrange multiplier $\lambda(\geq 0)$ is a parameter that controls how strongly the problem prefers parsimonious solutions: with a larger $\lambda$, the solution $\mathbf{s}^*$ for the optimization problem would tend to have fewer domains. Equivalently, a larger $\lambda$ prefers a larger-scale domain solution. Solving at multiple values of $\lambda$, therefore, may reveal the *multi-scale* domain structure in data.

In parallel to the statistical physics problem of a grand-canonical ensemble, $\mathcal{H}$ corresponds to the effective Hamiltonian of the system. In this view, $\lambda$ amounts to the negative *chemical potential* for adding extra domains to the solution. From the Bayesian viewpoint, on the other hand, our formulation is equivalent to considering a prior distribution of the form $p(\mathbf{s}) \propto \exp(-\lambda\mathcal{K}(\mathbf{s})/T)$, and a posterior distribution $p(\mathbf{s}|\mathbf{C}) \propto p(\mathbf{C}|\mathbf{s}) p(\mathbf{s}) \propto \exp(-\mathcal{H}_\lambda(\mathbf{s}|\mathbf{C})/T)$. The *maximum a posteriori* inference is equivalent to solving the minimization problem for $\mathcal{H}_\lambda(\mathbf{s}|\mathbf{C})$.

## Solving the inference problem

We solve the optimization problem at a fixed value of $\lambda$, to find the minimizer $\mathbf{s}^*$ for $\mathcal{H}_\lambda(\mathbf{s}|\mathbf{C})$. As the $\mathbf{s}$-space is expected to be high-dimensional and is likely characterized with multiple local minima, we use simulated annealing [71], in which the sampled distribution is narrowed down as $T$ is gradually decreased. At each value of $T$, we use a Markov chain Monte Carlo

sampling method to approximate the posterior distribution $p(\mathbf{s}|\mathbf{C}) \propto \exp(-\mathcal{H}_\lambda(\mathbf{s}|\mathbf{C})/T)$. See S3 Appendix for details.

We repeat the procedure to determine the best domain solutions at each different value of $\lambda$. This yields a family of multi-scale domain solutions, $\{\mathbf{s}_\lambda\}$. We note that there is no *a priori* notion of an optimal $\lambda$ at this point; instead, certain $\mathbf{s}_\lambda$'s may be more interesting or relevant, depending on the specific dataset and context. In Results, we applied our method to example Hi-C datasets, and identified and discussed the existence of such interesting solutions.

## Analysis and evaluation of domain solutions

**Similarity between two CD solutions.** To measure the extent of similarity between two CD solutions $\mathbf{s}$ and $\mathbf{s}'$, we evaluate the Pearson correlation. The binary matrices $\mathbf{B}$ and $\mathbf{B}'$ that represent the two CD solutions, are defined such that the matrix element are all 1's within the same CD and 0 otherwise. i.e., $(\mathbf{B})_{ij} = B_{ij} = \delta_{s_i s_j}$. The similarity between $\mathbf{B}$ and $\mathbf{B}'$ is quantified using the Pearson correlation

$$\rho = \frac{\langle \delta B \delta B' \rangle}{\sqrt{\langle (\delta B)^2 \rangle \langle (\delta B')^2 \rangle}}, \tag{18}$$

where $\langle \delta B \delta B' \rangle = \langle (B_{ij} - \bar{B})(B'_{ij} - \bar{B}') \rangle_{i \neq j}$, and $\langle (\delta B)^2 \rangle = \langle (B_{ij} - \bar{B})^2 \rangle_{i \neq j}$. The average $\langle \cdot \rangle_{i \neq j}$ runs over all distinct pairs.

**Normalized mutual information.** We use the mutual information to evaluate how well a CD solution $\mathbf{s}$ captures the visible patterns in the pairwise correlation data. We consider the binary grouping matrix $(\mathbf{B})_{ij} = B_{ij} = \delta_{s_i s_j}$ for the CD solution of interest, and compare it to the input data matrix $(\mathbf{A})_{ij} = A_{ij}$. In this study, either $\log_{10} \mathbf{M}$ or $\mathbf{C}_{O/E}$ was used for $\mathbf{A}$. Treating the matrix elements $a \in \mathbf{A}$ and $b \in \mathbf{B}$ as two random variables, we construct the joint distribution

$$p(A, B) = \langle \delta_{A_{ij}, a} \delta_{B_{ij}, b} \rangle_{i \neq j} \tag{19}$$

where $\langle \cdot \rangle_{i \neq j}$ is an average over all distinct pairs. The Kronecker delta for the continuous variable $a$ is defined in a discretized fashion: that is, $\delta_{A_{ij}, a} = 1$ if $A_{ij} \in [a, a + \Delta a)$ and 0 otherwise, where $\Delta a (= [\max\{A_{ij}\} - \min\{A_{ij}\}]/100)$ is used for discretization into 100 bins. Then we can calculate the mutual information,

$$I(A; B) = \sum_{a \in \mathbf{A}} \sum_{b \in \mathbf{B}} p(a, b) \log \left[ \frac{p(a, b)}{p(a)p(b)} \right], \tag{20}$$

and the normalized mutual information (nMI),

$$\text{nMI}(A; B) = \frac{I(A; B)}{\sqrt{H(A) \cdot H(B)}}, \tag{21}$$

where $H(X) = -\sum_{x \in \mathbf{X}} p(x) \log p(x)$ is the marginal entropy.

**Nestedness of CD solutions.** Here we define a measure to quantify the nestedness between two CD solutions, $\mathbf{s}$ (assumed to have smaller domains on average) and $\mathbf{s}'$ (larger domains). The idea is the following: $\mathbf{s}$ is perfectly nested in $\mathbf{s}'$ if, whenever two sites belong to a same domain in $\mathbf{s}$, they also belong to a same domain in $\mathbf{s}'$. For each domain $k \in \mathbf{s}$, we consider the best overlap of this domain $k$ on the other solution $\mathbf{s}'$:

$$h_1(k \to \mathbf{s}') = \max_{k' \in \mathbf{s}'} \frac{v_{k,k'}}{n_k} \tag{22}$$

where $v_{k,k'}$ is the number of overlapping sites between two domains $k \in \mathbf{s}$ and $k' \in \mathbf{s'}$, and $n_k$ is the size of domain $k$. The highest score $h_1(k \to \mathbf{s'}) = 1$ is obtained when domain $k$ is fully included in one of the domains in $\mathbf{s'}$. The null hypothesis corresponds to where the domains in $\mathbf{s}$ and $\mathbf{s'}$ are completely uncorrelated, in which case $h_1$ only reflects the overlap "by chance". The chance level $\bar{h}_1(k \to \mathbf{s'})$ is calculated by making $n_k$ random draws from $\mathbf{s'}$; we averaged over 100 independent trials. We normalize the score as

$$\hat{h}_1(k \to \mathbf{s'}) = \frac{h_1 - \bar{h}_1}{1 - \bar{h}_1}, \tag{23}$$

such that $\hat{h}_1(k \to \mathbf{s'}) = 0$ indicates the chance level, and $\hat{h}_1(k \to \mathbf{s'}) = 1$ means a perfect nestedness. Finally, we define the nestedness score $h(\mathbf{s} \to \mathbf{s'})$ for the entire CD solution as a weighted average:

$$h(\mathbf{s} \to \mathbf{s'}) = \sum_{k \in \mathbf{s}} \hat{h}_1(k \to \mathbf{s'}) \cdot \frac{n_k}{N}. \tag{24}$$

**The observed/expected matrix and its Pearson correlation matrix.** The observed/expected (O/E) matrix was used to account for the genomic distance-dependent contact number due to random polymer interactions in chromosome [18]. Each pair $(i, j)$ in O/E matrix is calculated by taking the count number $M_{ij}$ (observed number) and dividing it by average contacts within the same genomic distance $d = |i - j|$ (expected number). Since the expected number could be noisy, one smooths it out by increasing the window size (see refs. [18, 19] for further details). In this study, we used the expected number obtained from [19]. The Pearson correlation matrix of the O/E ($\mathbf{C}_{\mathrm{O/E}}$) represents the overall contact pattern through pairwise correlation coefficients between segments.

**Correlation between CTCF signal and domain boundaries.** The validity of domain boundaries, determined from various CD-identification methods including Multi-CD, is assessed in terms of their correlation with the CTCF signal. Suppose that the CTCF signal at genomic segment $i$ is given as $\phi_{\mathrm{CTCF}}(i)$. Then, we can consider an overlap function between $\phi_{\mathrm{CTCF}}(i)$ and a CD-boundary indicating function $\psi_{\mathrm{DB}}(i)$, where $\psi_{\mathrm{DB}}(i) = 1$ if the $i$-th segment is precisely at the domain boundary; $\psi_{\mathrm{DB}}(i) = 0$, otherwise. We evaluated a distance-dependent, normalized overlap function $\chi(d)$, defined as

$$\chi(d) = \frac{\langle \delta\phi_{\mathrm{CTCF}}(i + d)\psi_{\mathrm{DB}}(i)\rangle_i}{\langle \psi_{\mathrm{DB}}\rangle}, \tag{25}$$

where $\delta\phi_{\mathrm{CTCF}} = \phi_{\mathrm{CTCF}} - \langle\phi_{\mathrm{CTCF}}\rangle$. If the domain boundaries determined from Multi-CD is well correlated with TAD-capturing CTCF signal, a sharply peaked and large amplitude overlap function ($\chi(d)$) is expected at $d = 0$.

**Correlation between epigenetic marks and compartments.** We calculate the correlation of our compartment solutions with the epigenetic marks. Given a compartment solution $\mathbf{s}$ with two large domains A and B, we consider two binary vectors $\mathbf{q}^{(A)}$ and $\mathbf{q}^{(B)}$, where $q_i^{(A)} = +1$ if the $i$-th segment belongs to compartment A, and $q^{(A)} = -1$ otherwise. For a set of epigenetic marks measured across the genome is represented with $\mathbf{h}$, where its component $h_i$ denotes the value at the $i$-th genomic segment, the correlation between the solutions of

compartment *A* and *B* and the epigenetic marks can be evaluated using the Pearson correlations as:

$$c_A = \frac{(\mathbf{q}^{(A)} \cdot \mathbf{h})}{|\mathbf{q}^{(A)}||\mathbf{h}|}, \quad c_B = \frac{(\mathbf{q}^{(B)} \cdot \mathbf{h})}{|\mathbf{q}^{(B)}||\mathbf{h}|}. \tag{26}$$

## Data acquisition

All data used in the paper were obtained from publicly available repositories.

**Hi-C data.** We used Hi-C data from [19], available at NCBI GEO database [72] (https://www.ncbi.nlm.nih.gov/geo/), accession GSE63525. Analysis was performed using the intra-chromosomal contact matrices files for GSE63525- *celltype*, where *celltype* is replaced by one of the five cell type identifiers (GM12878-primary, HUVEC, NHEK, K562, and KBM7).

**Genomics data.** Information about genes on the human chromosome was obtained from the Known Gene table in the UCSC Genome Browser [73], and their annotated regulatory elements from the GeneHancer [61] interaction table, both accessed through the UCSC Table Browser [74] (https://genome.ucsc.edu/cgi-bin/hgTables). All data were consistent with the human genome assembly GRCh37 (hg19). Information about the APBB1IP gene was specifically obtained from the GeneCards database [75] (https://www.genecards.org).

**Biological markers.** The domain solutions from Multi-CD were compared with known biological markers. We obtained these data mostly from the ENCODE project [76]; the NCBI GEO [72] accession numbers are provided below. We used the enrichment data of the transcriptional repressor CTCF measured in a ChiP-seq assay from GEO accession GSM749704 (narrowpeak file). We binned the CTCF assay at 50-kb resolution, to match the Hi-C format. If there are multiple signal enrichments in a single bin, we took the average value. Because each CTCF signal has a finite width, there are occasional cases where a signal ranges across two bins; in those cases we evenly divided the signal strength into the two bins. The Repli-seq signals in the six phases G1, S1, S2, S3, S4, and G2 were obtained from GEO accession GSM923451, and the 11 histone mark signals from accession GSE29611. The Repli-seq and histone mark signals were averaged over 50-kb windows along the genome to construct the replication timing profiles. The RNA-seq data for the four cell lines GM12878, HUVEC, NHEK and K562 were obtained from GEO accession GSE33480. RNA-seq for the cell line KBM7 were separately obtained from [77] (https://opendata.cemm.at/barlowlab/2015_Kornienko_et_al/hg19/AK_KBM7_2_WT_SN.F.bw).

## Code availability

The Matlab software package and associated documentation are available online (https://github.com/multi-cd).

## Supporting information

**S1 Appendix. Gaussian polymer network for modeling chromosomes.**
(PDF)

**S2 Appendix. Derivation of the likelihood function.**
(PDF)

**S3 Appendix. The inference algorithm.**
(PDF)

**S1 Fig. Distance distributions of segment pairs are described by Gaussian.** (**A**) Gaussian probability distribution plotting $P(r_{ij})$ with different values of $\gamma_{ij}$ (Eq 2). The shaded area in different colors represents the corresponding values of contact probabilities, (Eq 5 at $r_c = 1$) (**B**) Distance distributions between one TAD (TAD17) and other TADs on Chr21 in human IMR90 cells measured with FISH. This figure was adapted from Fig 3A in [21]. (**C**) Distance distributions between three FISH probes on the X chromosome of male *Drosophila* embryos. The experimental data were digitized from Fig 3B in [42]. Their best fits to Eq 2 are plotted with solid lines. (**D**) Distance distributions between five pairs of FISH probes on chr1 in fibroblast cells. The experimental data (histograms) were digitized from Fig 4B in [43]. The fits using Eq 2 are plotted with solid lines. (**E**) Distance distributions between seven pairs of FISH probes in the Tsix/Xist region on the X chromosome of mouse ESC. The experimental data (black lines) were digitized from Fig 2F in [44], and their corresponding fits are shown in red. (PDF)

**S2 Fig. Chromatin domain solutions for chromosomes 4, 10, 11 and 19.** Extension of Fig 3. (**A**) Relative sizes of chromosomes considered, aligned at the centromeres. The gray shade in each chromosome indicates the 10-Mb interval for which we show the Hi-C data in the next panels. (**B-E**) Hi-C data for the corresponding 10-Mb genomic intervals of (**B**) chr4, (**C**) chr10, (**D**) chr11, and (**E**) chr19, for the five different cell lines respectively. All the panels for chr10 are reprints of Fig 3 in the main text. (**F-I**) Statistics of the domain solutions for chr4, chr10, chr11, and chr19. The five cell lines are color coded as indicated at the top of (**B**). (**F**) Mean domain size $\langle n \rangle$ as a function of λ. (**G**) The index of dispersion $D(= \sigma_n^2 / \langle n \rangle)$ of domain sizes. (**H**) The goodness of domain solutions, measured in terms of the normalized mutual information with respect to Hi-C data ($\log_{10}$ **M**). (**I**) The similarity of domain solutions across the five different cell types, measured by the Pearson correlation between binarized contact matrices. For each chromosome, arrows indicate the likely TAD scale (highest cell-to-cell similarity) and the likely meta-TAD scale (where the nMI is high and the index of dispersion $D$ starts to diverge). (PDF)

**S3 Fig. Examples of Multi-CD domain solutions at different scales.** Extension of Fig 3. Shown are the domain solutions obtained from Multi-CD for the five different cell lines (GM12878, HUVEC, NHEK, K562, KBM7), at (**A**) λ = 0 and (**B**) λ = 40. (PDF)

**S4 Fig. Scale-to-scale similarity of domain solutions for different cell lines.** Extension of Fig 3. We calculate the similarity between domain solutions at different λ in terms of Pearson correlation. The calculation was performed for chromosome 10 from five different cell lines. (PDF)

**S5 Fig. Link between chromatin organization and gene expression.** Extension of Fig 3K (gene APBB1IP, reproduced in the left panel), with two additional genes SVIL (middle panel) and TSPAN15 (right panel). Top five rows: Colored blocks indicate the TAD solutions for the five cell lines, as identified from our method. Colored hairy lines show the RNA-seq signal for the respective cell lines. Bottom two rows: Black horizontal bar and the gray shade mark the known position range of the gene. Finally, positions of all known regulatory elements for the gene are shown, as annotated in the GeneHancer database [61]. (PDF)

**S6 Fig. Identification of sub-TAD boundaries at 5-kb resolution.** (**A**) The optimum cluster size, best describing 5-kb resolution Hi-C map in terms of nMI, is determined at

$\langle n \rangle$ = 0.35 Mb, which is consistent with the sub-TAD size determined from 50-kb resolution Hi-C at $\lambda = 0$. (**B-C**) Comparison between Multi-CD solutions at different resolutions of the input Hi-C data, that point to the robustness of sub-TAD boundaries regardless of Hi-C resolution. (**B**) The best CD solution (corresponding to $\lambda = \lambda^*$ in panel (**A**)) for the 5-kb resolution Hi-C data in the 120-124 Mb region of the genome. (**C**) Solution for the same genomic interval from 50-kb Hi-C, determined at $\lambda = 0$. The two CD solutions are effectively identical, which supports our interpretation of sub-TAD as the unit of hierarchical chromosome organization.
(PDF)

**S7 Fig. Comparison of domain solutions from Multi-CD and other methods at specific scales.** Comparison between domain solutions obtained by three popular algorithms (Arrow-Head, DomainCaller, GaussianHMM) (right column) and those by Multi-CD (left column), applied to 50-kb resolution Hi-C data. Three subsets from the same Hi-C data ($\log_{10} \mathbf{M}$), with different magnification (5, 10, and 40 Mb from top to bottom), are given in the middle column. ArrowHead algorithm [19] was used for identifying the domain structures of sub-TADs, DomainCaller [22] for TADs, and Gaussian Hidden Markov Model (GaussianHMM) [19] for compartments. Multi-CD use $\lambda = 0, 10, 90$, as the parameter values for identifying sub-TADs, TADs, and compartments, respectively.
(PDF)

**S8 Fig. Comparison of histone marks and compartments.** Extension of Fig 6C and 6D, which make comparison between the CD solutions for A/B-compartments by Multi-CD and epigenetic marks. The upper part with Repli-Seq signals is a reprint from the main text figure. The lower part shows histone marks on the corresponding genomic range. Majority of the histone marks are correlated with the A-compartment. The values of Pearson correlation between Repli-Seq signal or histone marks and A/B-compartment are given on the right.
(PDF)

**S9 Fig. Comparison to existing algorithms for identifying domains at multiple scales.** (**A**, **B**) Normalized mutual information between domain solutions at multiple scales, from Multi-CD, Armatus [34] and TADtree [33] respectively, and the log10 of KR-normalized Hi-C matrix for chr10 of the cell line GM12878. The scale of a domain solution $\mathbf{s}$ is measured in two ways, in terms of (**A**) the effective number of clusters, $\mathcal{K}(\mathbf{s}) = \exp(-\sum_{k=1}^{K}(n_k/N)\log(n_k/N))$, where $n_k = \sum_{i=1}^{N}\delta_{s_i,k}$ is the domain size; and (**B**) the total area of 1's in the corresponding binary contact matrix, $(\text{area}) = \sum_{i,j=1}^{N} B_{ij}$ where $B_{ij} = \delta_{s_i,s_j}$. All domain solutions from TADtree and Armatus were obtained using the respective default parameter settings. (**C-F**) Visual comparison of domains found by (**C**, **D**) TADtree and Multi-CD, and (**E**, **F**) Armatus and Multi-CD, at matching scales in terms of the average domain size. Domain solutions are shown in the upper triangle, colored by red (intra-domain) and white (extra-domain) for effective visualization. The lower triangle plots the corresponding subset of the Hi-C data (KR-normalized and in log10). Refer to the original papers [33, 34] for the definitions of the respective control parameters $\alpha$ (TADtree) and $\gamma$ (Armatus).
(PDF)

## Acknowledgments

We thank Roger Oria Fernandez for feedback on the code. We thank the Center for Advanced Computation in KIAS for providing computing resources.

## Author Contributions

**Conceptualization:** Ji Hyun Bak, Min Hyeok Kim, Changbong Hyeon.

**Data curation:** Ji Hyun Bak.

**Formal analysis:** Ji Hyun Bak, Min Hyeok Kim.

**Funding acquisition:** Changbong Hyeon.

**Investigation:** Ji Hyun Bak, Min Hyeok Kim, Lei Liu, Changbong Hyeon.

**Methodology:** Ji Hyun Bak, Min Hyeok Kim, Lei Liu, Changbong Hyeon.

**Project administration:** Changbong Hyeon.

**Resources:** Changbong Hyeon.

**Software:** Ji Hyun Bak, Min Hyeok Kim.

**Supervision:** Changbong Hyeon.

**Validation:** Ji Hyun Bak, Min Hyeok Kim.

**Visualization:** Ji Hyun Bak, Min Hyeok Kim.

**Writing – original draft:** Ji Hyun Bak, Min Hyeok Kim, Lei Liu, Changbong Hyeon.

**Writing – review & editing:** Ji Hyun Bak, Changbong Hyeon.

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
