## [Decision Letter · Decision Letter 0]

25 Nov 2020

Dear Prof. Hyeon,

Thank you very much for submitting your manuscript "A unified framework for inferring the multi-scale organization of chromatin domains from Hi-C" for consideration at PLOS Computational Biology.

As with all papers reviewed by the journal, your manuscript was reviewed by members of the editorial board and by several independent reviewers. In light of the reviews (below this email), we would like to invite the resubmission of a significantly-revised version that takes into account the reviewers' comments.

We cannot make any decision about publication until we have seen the revised manuscript and your response to the reviewers' comments. Your revised manuscript is also likely to be sent to reviewers for further evaluation.

Sincerely,

Alexandre V. Morozov, Ph.D.

Associate Editor

PLOS Computational Biology

Jian Ma

Deputy Editor

PLOS Computational Biology

Reviewer's Responses to Questions

**Comments to the Authors:**

Reviewer #1: The authors developed a unified method for analyzing Hi-C data and determine multi-scale organization of chromatin chain by changing the parameter λ. The method was applied to several cell lines and obtained valuable results that are consistent with the experimental observations. The methods and the results are quite interesting. I believe it is suitable for publication after the revision. There are some issues need to be addressed:

1. The proposed method and the model are fully depended on the probability, which is obtained from the normalized Hi-C data. There are several normalization methods for Hi-C data, including ICE, KR, VC, etc. The authors should talk about the influence of the normalization algorithm on the results of the Sub-TADs, TADs, meta-TADs, and the compartments.

2. The authors claimed that the CD solutions correspond to the compartments at about λ=90. How to determine this value? If λ changes to 80 or 100, the distribution of the compartments stays unchanged or changes significantly? In addition, the results of large λ values (70 to 100) are not shown in Fig. 3.

3. The analysis of the link between chromatin organization and gene expression is very interesting and meaningful. The authors mentioned that the domain conservation was strongest at λ=10. Is this value for the results of GM12878 chr10, or the other four cell lines have the same value? In addition, can the authors add the analysis of two other gene expressions that support their results (Fig. 3k)?

4. Only part of the short arm of chr10 is shown in this study. Did the authors construct the model for the whole chr10 and calculate the properties, or for only a part of chr10? The component of the model is not so clearly described in this study. I think the results of parameter λ will change a lot for different cells, chromosomes, and regions (Fig. S2).

Reviewer #2: The authors developed a Hi-C data analysis pipeline, called Multi-CD, including two steps: pre-processing and inference based on a polymer physics approach and a statistical physics sampling approach. Interestingly, they formulated the inference problem with a single parameter λ relating to multi-scale chromatin domains and tried to provide a unified framework into the hierarchical chromatin organization. Significantly, the idea of combining the correlation matrix and group modeling is very original and innovative in the field of 3D genome physical biology. As shown in Figure 6, the outcomes by this method are consistent with other reported methods. Besides, the hierarchical organization of chromatin domain families in Figure 5 clearly shows that their unified framework works well. However, this manuscript includes little biological insights. The reviewer suggests that this methodological framework is worth publishing as not a research paper but a method one. Before a final assessment, the authors need to address the following comments in a revised manuscript.

Major:

1) at l399: γij has a physical unit [m^-2]. Therefore, stiffness or the spring constant is not the correct expression.

2) at l441: It is not clear how to solve Equation (4) from pij to γij in this method, although the uniqueness might be confirmed.

3) at l444 and Equation (5): the assumption σii = σjj = σc = median(1/4γij) is too arbitrary. The reviewer guesses that the authors could not derive the one-to-one connection between γij and σij.

According to a rigorous theory of the Gaussian polymer network [55 and https://doi.org/10.1016/j.csbj.2020.08.014], the Kirchhoff or Laplacian matrix is positive semidefinite, and the smallest eigenvalue is 0 with the eigenvector proportional to (1, 1, ..., 1). Therefore, an inverse matrix does not exist. On the other hand, γij^-1 = 2(σii + σjj - 2σij) at l405 is a rigorous relationship. Besides, the matrix Σ=(σij) satisfies Σ (1, 1, ..., 1)^T = 0. Using this relation, we can derive (γij^{-1}) (1, 1, ..., 1)^T = 2 A (σ11, σ22, ..., σNN)^T, where the diagonal elements of the matrix A are N+1 and the non-diagonal elements are 1. The inverse matrix of A is \\frac{1}{2N^2} B, where the diagonal elements of the matrix B are 2N-1 and the non-diagonal elements are -1. Therefore, we can derive the diagonal elements (σ11, σ22, ..., σNN) only from the matrix (γij^{-1}) and solve the equation γij^-1 = 2(σii + σjj - 2σij) without the assumption σii = σjj = σc = median(1/4γij).

4) Although the reviewer cannot verify the Matlab codes, information regarding the computational calculation time to obtain an optimal solution for a parameter λ must be useful for PLOS CB readers.

5) In terms of bioinformatics and computational approaches, the parameter λ would be an excellent measure to identify multi-scale chromatin domains. Also, the authors have discussed the interpretation of λ as the negative chemical potential. However, if the authors are responsible for the meaning of λ in terms of physics, they should discuss and propose a way to find a physical meaning of λ by experiments.

Minor:

1) typo at l76: the presence "of" of cell-to-cell ... Remove "of."

2) at l396: Equation (2) does not represent the distance distribution but the probability density function of the distance.

**Have all data underlying the figures and results presented in the manuscript been provided?**

Reviewer #1: Yes

Reviewer #2: Yes

PLOS authors have the option to publish the peer review history of their article (what does this mean?). If published, this will include your full peer review and any attached files.

Reviewer #1: No

Reviewer #2: No
---

## [Decision Letter · Decision Letter 1]

8 Feb 2021

Dear Prof. Hyeon,

Thank you very much for submitting your manuscript "A unified framework for inferring the multi-scale organization of chromatin domains from Hi-C" for consideration at PLOS Computational Biology. As with all papers reviewed by the journal, your manuscript was reviewed by members of the editorial board and by several independent reviewers. The reviewers appreciated the attention to an important topic. Based on the reviews, we are likely to accept this manuscript for publication, providing that you modify the manuscript according to the review recommendations.

Sincerely,

Alexandre V. Morozov, Ph.D.

Associate Editor

PLOS Computational Biology

Jian Ma

Deputy Editor

PLOS Computational Biology

[LINK]

Reviewer's Responses to Questions

**Comments to the Authors:**

Reviewer #1: I believe that the authors have already addressed all my concerns. Therefore I have no further comments and recommend its acceptance.

Reviewer #2: The authors gave answers to my comments and improved the quality of the manuscript significantly. However, some of the points they addressed are still not clear. I believe, after addressing these concerns properly, this manuscript can be accepted to PLOS Computational Biology.

Comment 1 to the reply R2-3:

I understood that the authors used the tar formatted Hi-C matrix data, and the diagonal elements were missing in the format.

But, if one uses the hic formatted data, the diagonal elements are available. Moreover, invalid pairs, including self-ligation, are eliminated in a Hi-C analysis pipeline from FASTQ data. Therefore, the diagonal element p_ii indicates not self-ligation counts but the contact frequency within the i-th genomic region.

I am looking forward to your future work with improving these parts.

Comment 2 to the reply R2-5:

The authors should add a discussion on the physical implication of λ.

**Have all data underlying the figures and results presented in the manuscript been provided?**

Reviewer #1: Yes

Reviewer #2: Yes

PLOS authors have the option to publish the peer review history of their article (what does this mean?). If published, this will include your full peer review and any attached files.

Reviewer #1: No

Reviewer #2: No
---

## [Editor Report · Decision Letter 2]

23 Feb 2021

Dear Prof. Hyeon,

We are pleased to inform you that your manuscript 'A unified framework for inferring the multi-scale organization of chromatin domains from Hi-C' has been provisionally accepted for publication in PLOS Computational Biology.

Best regards,

Alexandre V. Morozov, Ph.D.

Associate Editor

PLOS Computational Biology

Jian Ma

Deputy Editor

PLOS Computational Biology

---

## [Editor Report · Acceptance letter]

10 Mar 2021

PCOMPBIOL-D-20-01936R2 

A unified framework for inferring the multi-scale organization of chromatin domains from Hi-C

Dear Dr Hyeon,

I am pleased to inform you that your manuscript has been formally accepted for publication in PLOS Computational Biology. Your manuscript is now with our production department and you will be notified of the publication date in due course.

With kind regards,

Alice Ellingham
